# Comparisons of GC-Measured Carboxylic Acids and AMS m/z 44 Signals: Contributions of Organic Acids to m/z 44 Signals in Remote Aerosols from Okinawa Island

**Bhagawati Kunwar** [1,2] , **Kazuhiro Torii** [1,3], **Shankar G. Aggarwal** [1,4], **Akinori Takami** [5]
**and Kimitaka Kawamura** [1,2,*]

1    Institute of Low Temperature Science, Hokkaido University, N19 W8, Kita-ku, Sapporo 060-0819, Japan
2    Chubu Institute for Advanced Studies, Chubu University, 1200 Matsumoto-cho, Kasugai 487-8501, Japan
3    Graduate School of Environmental Science, Hokkaido University, N11 W5, Kita-ku, Sapporo 060-0810, Japan
4    CSIR-National Physical Laboratory, New Delhi 110012, India
5    Center for Regional Environmental Research, National Institute for Environmental Studies,
     Tsukuba 305-8506, Japan
*    Correspondence: kkawamura@isc.chubu.ac.jp

**Highlights:**

- LMW dicarboxylic acids determined by GC explain ca. 50% of AMS-derived m/z 44 signals
- The remaining half may be contributed by LMW monocarboxylic acids in the aerosol phase.
- Organic aerosols from the western North Pacific Rim are photochemically more aged.

**Abstract:** An intercomparison study was conducted to evaluate the contributions of carboxylic acids to m/z 44 ($COO^+$) signals obtained by an on-line aerosol mass spectrometer (AMS) during a field campaign at Cape Hedo, Okinawa, in the western North Pacific Rim. We report for the first time that carboxylic acids (diacids, oxoacids, benzoic acid, and fatty acids) significantly contribute to m/z 44 signals with a strong correlation (R = 0.93); oxalic acid accounts for $16 \pm 3\%$ of the m/z 44 signals and $3.7 \pm 0.9\%$ of organic mass measured by AMS. We also found that about half of AMS m/z 44 signals can be explained by diacids and related compounds, suggesting that the remaining signals may be derived from other organic acids including monocarboxylic acids (e.g., formate and acetate) in aerosol phase. This study confirms that AMS-derived m/z 44 can be used as a surrogate tracer of carboxylic acids, although the signals cannot specify the types of carboxylic acids and their molecular compositions.

**Keywords:** carboxylic acids by GC; m/z 44 signals by AMS; comparison between GC and AMS; photochemical processing; long-range atmospheric transport

## 1. Introduction

Organic aerosols (OA) are a ubiquitous and abundant component in the atmosphere. They account for 20–50% of the fine particle mass [1–4] and can be primarily formed by fossil fuel combustion and biomass burning [5] and secondarily by gas-to-particle conversion of volatile organic compounds (VOCs) via photochemical processing with $O_3$ and OH radicals [6–10]. In both continental and marine aerosols, oxalic acid has been reported as the most abundant dicarboxylic acid followed by malonic and/or succinic acid using gas chromatography (GC) and the GC/mass spectrometer [11–13].

Low-molecular-weight (LMW) dicarboxylic acids (hereafter, diacids) and other organic acids are ubiquitous in the atmosphere, with high concentrations up to few μg m$^{-3}$ [14–19], demonstrating that they are the most important constituents of tropospheric OA [13,20]. Diacids and related compounds can contribute to 0.2–1.8% of total carbon (TC) in urban aerosols from Tokyo [14] and up to 16% of TC in remote marine aerosols from the central

Pacific, including the tropics [11]. Previous studies demonstrated that diacids account for up to 10–27% of organic carbon (OC) in aerosols, e.g., [13,21,22].

Although air pollutants in East Asia have largely increased in recent decades with a significant influence over the continents and ocean on regional and global scales [23], levels of some pollutants such as sulfate started to decline recently in Chinese megacities and outflow regions [24,25], except for surface ozone [26] and water-soluble dicarboxylic acids [27]. Organic aerosols emitted from East Asia are transported to the western North Pacific Rim, including Okinawa Island [28]. Irei et al. [29] utilized an aerosol mass spectrometer (AMS) at Cape Hedo, Okinawa, to study the ambient aerosols influenced from East Asia and reported a good relation between AMS m/z 44 and water-soluble organic carbon (WSOC). The AMS studies for organic aerosols have been reported from many field campaigns [30–37]. The AMS signal of m/z 44, mainly originating from $CO_2^+$, have been used as a marker for oxygenated organics in urban areas [38–41].

However, direct comparison is lacking between AMS-derived m/z 44 ($COO^+$) signals and GC-measured carboxylic acids in aerosols, except one comparison reported in urban Tokyo [40]. Although there are many studies about m/z 44 ($COO^+$) by AMS [29,39,41–47], a direct comparison of various types of carboxylic acids determined by GC/FID with on-line AMS measurement has not been conducted from remote oceanic sites.

In this study, we measured various types of carboxylic acids, including diacids, oxoacids, and fatty acids, in coastal aerosols collected from Cape Hedo, Okinawa Island, using butyl ester derivatization for off-line GC determination. AMS measurements were also conducted on-line at the same site during the same campaign. The concentrations of those acids were compared with those of m/z 44 ($COO^+$) signals obtained by on-line AMS at the same site. Here, we compare, for the first time, the GC/FID results of various dicarboxylic acids and related compounds with AMS-derived m/z 44 signals for the marine aerosol particles from the western North Pacific Rim.

## 2. Samples and Analytical Procedure

### 2.1. Site Description and Aerosol Sampling

Aerosol samples ($PM_{1.0}$, *n* = 28) were collected during a field campaign from 17 March 2008 to 13 April 2008 using low volume air sampler and pre-combusted (450 °C, 4 h) quartz fiber filters (Pallflex 2500QAT, 47 mm in diameter). The sampler was installed on the roof top of the first story of the Cape Hedo Atmosphere and Aerosol Monitoring Station building (CHAAMS, 26°9′ N, 128°2′ E) [48], which is located at Cape Hedo in the northwest coast of Okinawa Island, Japan, an outflow region of East Asia (Figure 1). The flow rate was 16.7 L/min and sample volume is 24 $m^3$. Each sample was collected for 24 h. Before and after sampling, the filters were stored in a preheated glass vial (50 mL) with a Teflon-lined screw cap. After the collection, samples were stored in a freezer at −20 °C to prevent the microbial degradation of organics. More details are described elsewhere [28,49]. Possible artifacts on the filters due to the evaporation and adsorption of semivolatile organic compounds during sample collection could not be eliminated.

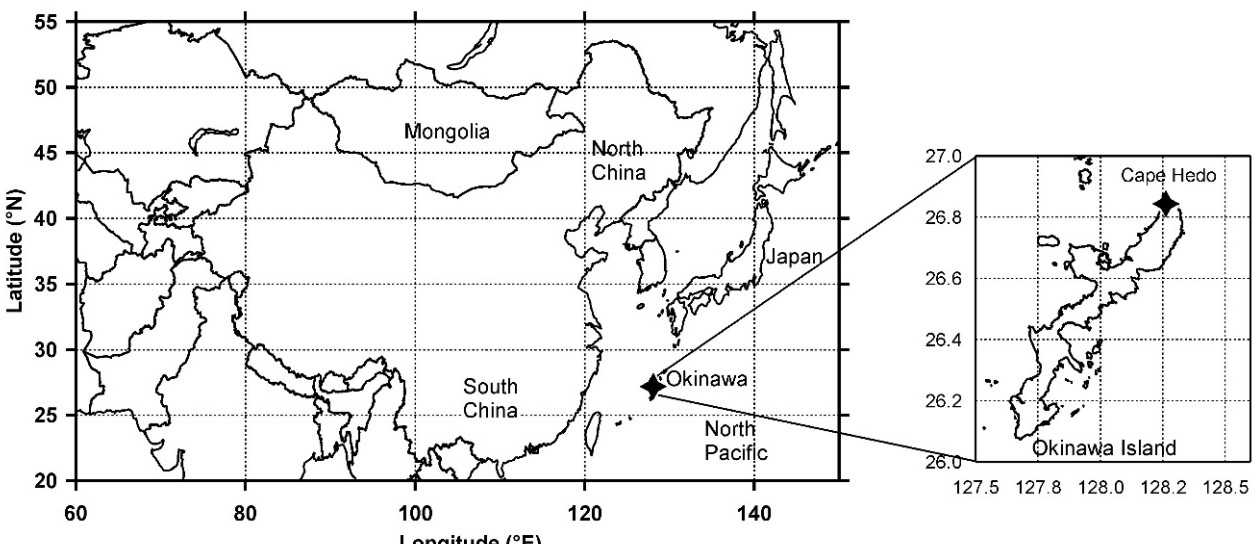

**Figure 1.** Map of Okinawa Island, Japan, with sampling location (Cape Hedo) for aerosols ($PM_{1.0}$) and on-line AMS measurements.

### 2.2. Chemical Analysis

Filter samples ($PM_{1.0}$) were analysed for water-soluble diacids, oxoacids, and fatty acids by the method reported previously [7,50]. Briefly, the known area of the filter was extracted with organic free pure water, and then organic acids (diacids, oxoacids, benzoic acid, and fatty acids) were derivatized with 14% $BF_3$/n-butanol to derive dibutyl esters, dibutoxy acetal/butyl esters, and butyl esters. These derivatives were determined using a capillary GC (HP 6890) equipped with a split/splitless injector, a HP-5 fused silica capillary column (0.2 mm i.d. × 25 m long × 0.50 μm film thickness), and a flame ionization detector (FID). The GC peak areas for the derivatives were calculated with a Shimazdu C-R7Aplus data system. The GC peaks were identified by comparing GC retention times with authentic standards and confirmed by GC/mass spectrometry (GC/MS). The concentrations of all the species reported here are corrected for field blanks. We also performed the recovery test by spiking authentic diacids to the quartz filter and analyzing it as we would a real sample. The recoveries were 90% for oxalic acid ($C_2$) and more than 95% for malonic acid ($C_3$) to azelaic acid ($C_9$). The analytical error of $C_2$ is less than 2% and those of other species are less than 5%.

### 2.3. Aerodyne AMS Measurements

Aerodyne quadrupole aerosol mass spectrometer (Q-AMS; Aerodyne Research, Inc., Boston, MA, USA) was deployed for the springtime campaign at Cape Hedo, Okinawa, with a time resolution of 10 min [48,51]. A quadruple mass spectrometer was utilized to analyze the positive ions for unit mass-to-charge (m/z) ratios with a mass resolution of 200. Setup conditions for ambient measurement were the same as for the previous observation at Fukue [52]. A detailed description of the Q-AMS can be found elsewhere [48,51,53–55]. Briefly, aerosols are separated from gaseous species by an aerodynamic lens and vaporized at 600 °C on a vaporizer. The size cut of the aerodynamic lens is approximately $PM_{1.0}$ [51]. The vaporized molecules were then ionized by electron impact at 70 eV, and positive ions were measured by a quadrupole mass spectrometer, which gives the mass spectra of aerosol components [56]. AMS m/z 44 ($CO_2$) signals mainly originate from the decarboxylation of organic acids that are vaporized at 600 °C prior to ionization [40,41,46,57,58]. AMS data

were averaged for the integration time of filter samples, i.e., 24 h. Quantified data by AMS were converted to organic aerosol masses using the following equations [51].

$$M_{\mathrm{m/z}} = \frac{1}{\overline{CE}_{org}} \frac{1}{\overline{RIE}_{org}} \frac{MW_{\mathrm{NO3}}}{IE_{\mathrm{NO3}}} \frac{10^{12}}{QN_A} S_{\mathrm{m/z}} \tag{1}$$

$$OA = \sum_{\mathrm{m/z}} M_{\mathrm{m/z}}(\mathrm{m/z} = 1 - 300) \tag{2}$$

where $\overline{CE}_{org}$ and $\overline{RIE}_{org}$ denote the average particle collection efficiency and relative ionization efficiency (RIE) for organics, respectively.

Similarly, $MW_{\mathrm{NO3}}$ (62 g mol$^{-1}$) indicates the molecular weight of nitrate, whereas $IE_{\mathrm{NO3}}$ indicates ionization efficiency of ammonium nitrate. Q denotes a sample flow rate in cm$^3$S$^{-1}$, and N$_A$ denotes Avogadro's number. $S_{\mathrm{m/z}}$ (Hz) is the signal of count rate at the m/z originating from organic compounds and obtained by the subtraction in the signals from the ambient gas molecules, inorganic species, and instrumental artifacts. On the other hand, $IE_{\mathrm{NO3}}$ represents the determined monodisperse ammonium nitrate particles from the calibration unit. We assume that the $\overline{RIE}_{org}$ is 1.4 based on the work of Alfarra et al. [59], and the CE value is assumed to be 0.5 for organics following the study of Takegawa et al. [40], who used authentic diacid standards to estimate the CE value. Takegawa et al. [40] reported that the size distribution of organics in Tokyo is bimodal. Similar case is also reported for AMS measurements in urban areas, e.g., [51,54,55,60]. The small mode (d$_{va}$ < 200 nm) is dominated by aliphatic or aromatic hydrocarbons mainly emitted primarily from combustion sources. The accumulation mode (d$_{va}$ > 200 nm) is dominated by oxygenated organic compounds.

The oxygenated organic compounds are the main constituents of secondary organic aerosols. The small-mode organics are externally mixed with inorganic species, and the accumulation-mode organics are internally mixed with inorganic species, e.g., [55,60]. Therefore, the AMS particle collection efficiency of 0.5 is reasonable for organics in the study site. CE is actually defined for particles and not species and in principle could be a function of particle size for the same composition and physical shape [59]. CE values were determined by comparing the AMS mass loading for a given chemical species with mass loading from other particulate chemical measurement techniques such as the particle-into-liquid sampler (PILS) with ion chromatography [40,61], the AMS total mass loading (sum of all chemical species) with other numbers, and mass-based instrument techniques, such as scanning mobility particle sizer (SMPS) [62] and tapered element oscillating microbalance (TEOM) [63].

Several earlier field campaigns [64–67] showed that the mass of sulfate detected by the AMS was often low by a factor of 1.5–2.3, implying an apparent CE value for sulfate between 0.4 and 0.7. CE is also affected by relative humidity (RH) [64], particle acidity (Quinn et al., 2006 [62]), and nitrate (Crosier et al., 2007 [68]). Other laboratory studies of pure organic species reported that the CE depends on the phase of the particles, with a CE of 1 for liquids and 0.2–0.5 for solids [59]. Zhang et al. [42] used an average collection efficiency of 0.7 for total organics in non-refractory-PM$_{1.0}$ (CE = 1 for small-mode organics and CE = 0.5 for accumulation-mode organics), based on laboratory experiments performed by Slowik et al. [69].

AMS can quantify both organic and inorganic species using a flash vaporization followed by electron impact ionization. The thermal decomposition of organic acids to CO$_2$ is well known, and m/z 44 fragment can be formed by the decarboxylation of mono-, di-, and poly-carboxylic acids [40,59,70,71]. Takegawa et al. [40] reported the mass fragmentation patterns for major organic acids using authentic standards. For oxalic acid, m/z 44 peak is the largest, accounting for ~34% of the sum of the fragment ions. The values of malonic (C$_3$) and succinic (C$_4$) acids were 21% and 15%, respectively. On the other hand, those of glutaric (C$_5$), adipic (C$_6$), glyoxylic ($\omega$C$_2$), and phthalic (Ph) acids were 5–7% of

the sum of all the fragments. However, other diacids and oxoacids and fatty acids as well as salt forms of organic acids were not examined in Takegawa et al. [40].

## 3. Results and Discussions

### 3.1. Molecular Distributions and Temporal Variation in Dicarboxylic Acids, Oxoacids, Benzoic Acid, and Fatty Acids

We summarize the concentrations of organics: organic carbon and diacids, oxoacids, benzoic acid, and fatty acids obtained (Table S1). Carboxylic acids and OC were obtained by the GC and carbon analyzer, respectively. A homologous series of $\alpha,\omega$-dicarboxylic acids ($C_2$–$C_{12}$), unsaturated diacids (phthalic, isophthalic, terephthalic, maleic, fumaric and methylmaleic), multifunctional diacids (malic, ketomalonic, and 4-ketopimelic), oxocarboxylic acids ($\omega C_2$–$\omega C_9$ and pyruvic acid), and $\alpha$-dicarbonyls (glyoxal and methylglyoxal), benzoic acid and fatty acids was detected in the $PM_{1.0}$ samples collected during the campaign.

Oxalic acid ($C_2$) was found as the most abundant organic species followed by malonic ($C_3$), succinic ($C_4$), glyoxylic ($\omega C_2$), ketomalonic ($kC_3$), and phthalic (Ph) acids. The predominances of $C_2$ in diacids and $\omega C_2$ in oxoacids indicate a significant photochemical oxidation of organic precursors in both gaseous and aqueous phases during long-range atmospheric transport [13,49,72–77]. Their predominance is due to secondary formation from both biogenic and anthropogenic sources. 9-Oxononanoic ($\omega C_9$) is produced by the photo-oxidation of biogenic unsaturated fatty acids such as oleic acid [78]. Benzoic acid is emitted from automobile exhausts [79]. Kunwar et al. [75] reported that a secondary formation is the dominant source of organic acids in Okinawa from their precursors.

Total diacids are defined as the sum of all the diacids detected. The relative abundance of $C_2$ is defined as the $C_2$ (%) of total diacid concentration; $C_2$ (%) = (concentration of $C_2$/concentration of total diacids) × 100. During the campaign, the average $C_2$ (%) was 75%. The sum of major diacids ($C_2$, $C_3$, $C_4$ and $kC_7$) comprised 96.4% of $\Sigma C_2$–$C_{12}$ diacids (summation of $C_2$ to $C_{12}$) (not shown as a figure). Similarly, the relative abundance of $\omega C_2$ in total oxoacids was 81%. The sum of $\omega C_2$ and $\omega C_9$ comprised 90% of total oxoacids. $C_2$, $C_3$, $C_4$, $kC_7$, $\omega C_2$, $\omega C_9$, and benzoic acid are major carboxylic acids that constitute the molecular compositions of diacids and related compounds. We calculated the acid mass concentration according to Yatavelli et al. [41]. The acid mass fraction for selected diacids and related compounds (concentrations of diacids, oxoacids, benzoic acid, pyruvic acid, and fatty acids/$Org_{AMS}$) was 0.08, whereas the ratio of m/z 44/$Org_{AMS}$ was 0.12. The fraction of carboxylic acids obtained from GC measurement was 1.5 times lower than that from AMS m/z 44, suggesting the presence of other organic acids including monoacids such as formic, acetic, and propanoic acids. Interestingly, high abundances of formic and acetic acids were reported in gas and particle phases from Mt. Tai in the North China Plain [80].

We selected total diacids, total oxoacids, benzoic acid, and total fatty acids to compare with AMS m/z 44 signals (Table S1). Figure 2 shows the average relative abundances of total diacids, total oxoacids, benzoic acid, and total fatty acids in total carboxylic acids. The average relative abundance of total diacids in total carboxylic acids is 70.7%, followed by total fatty acids (18%), total oxoacids (7%), and benzoic acid (4.4%). The higher % of diacids suggests that dominant organics are formed from secondary formation. We consider that the above-mentioned carboxylic acids can potentially contribute to the m/z 44 signals obtained by AMS [40]. Using the $CO^+$/$CO_2^+$ ratio of unity [40], we calculated the potential contribution of oxalic acid to m/z 44 signals (see Text S1 in Supplementary Materials).

We found that oxalic acid contributes to 16 ± 3.5% of the m/z 44 signal, which is equivalent to 3.45 ± 0.9% of organic matter measured by AMS ($OM_{AMS}$). The rest of the m/z 44 signal should originate from other diacids as well as mono-/poly-carboxylic acids. Chen et al. [81] reported a significant amount of low-molecular-weight monocarboxylic acids using high-resolution Time-of-Flight AMS in the Southeastern U.S.

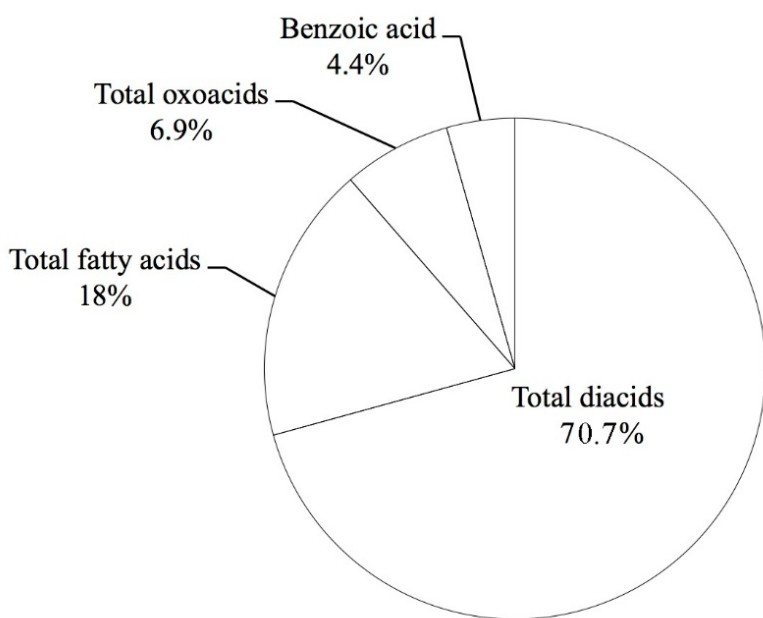

**Figure 2.** Averaged relative abundances of total diacids, total oxoacids ($\omega C_2$–$\omega C_9$ and pyruvic acid), benzoic acid, and fatty acids ($C_7$–$C_{20}$) in total carboxylic acids detected in the Cape Hedo aerosols ($PM_{1.0}$).

### 3.2. Temporal Variations and Comparison of m/z 44 Signal with COO of Total Carboxylic Acids

Figure 3 shows the temporal variations in $OM_{AMS}$ and m/z 44 ($COO^+$) signals. The real-time variations in $OM_{AMS}$ and m/z 44 can be found in Figure S1. We found a very strong correlation between $OM_{AMS}$ and m/z 44 signal (R = 0.98). Figure 4 shows the temporal variations in the estimated COO for selected carboxylic acids together with AMS m/z 44 signals. $C_2$-$C_4$ diacids $\omega C_2$, $\omega C_4$, and $\omega C_9$ are mainly produced by the photochemical oxidation of various organic precursors during long-range atmospheric transport to Okinawa [28,49]. The temporal variations in COO calculated for total carboxylic acids are similar to those of m/z 44. The m/z 44 signals mainly originate from oxygenated organics [39,42]. Although the /z 44 peak is the largest in AMS mass spectrum, $C_3H_8^+$ and $C_2H_4O^+$ may also contribute to the m/z 44 signals [41]. Some amino compounds may produce significant peaks at m/z 44 ($C_2H_6N^+$) and m/z 28 ($CH_2N^+$). However, these compounds are unlikely to have contributed to those signals in this study because the mass spectra do not show the presence of $C_nH_{2n+1}NH^+$ ion series, which are characteristic of alkylamines [42,54].

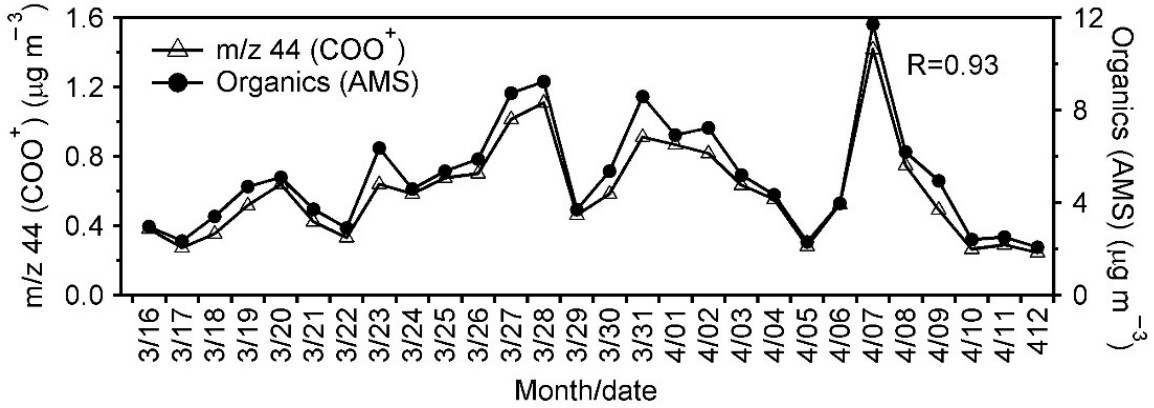

**Figure 3.** Temporal variations in organics and m/z 44 ($COO^+$) signal measured by AMS in aerosols samples collected from Okinawa.

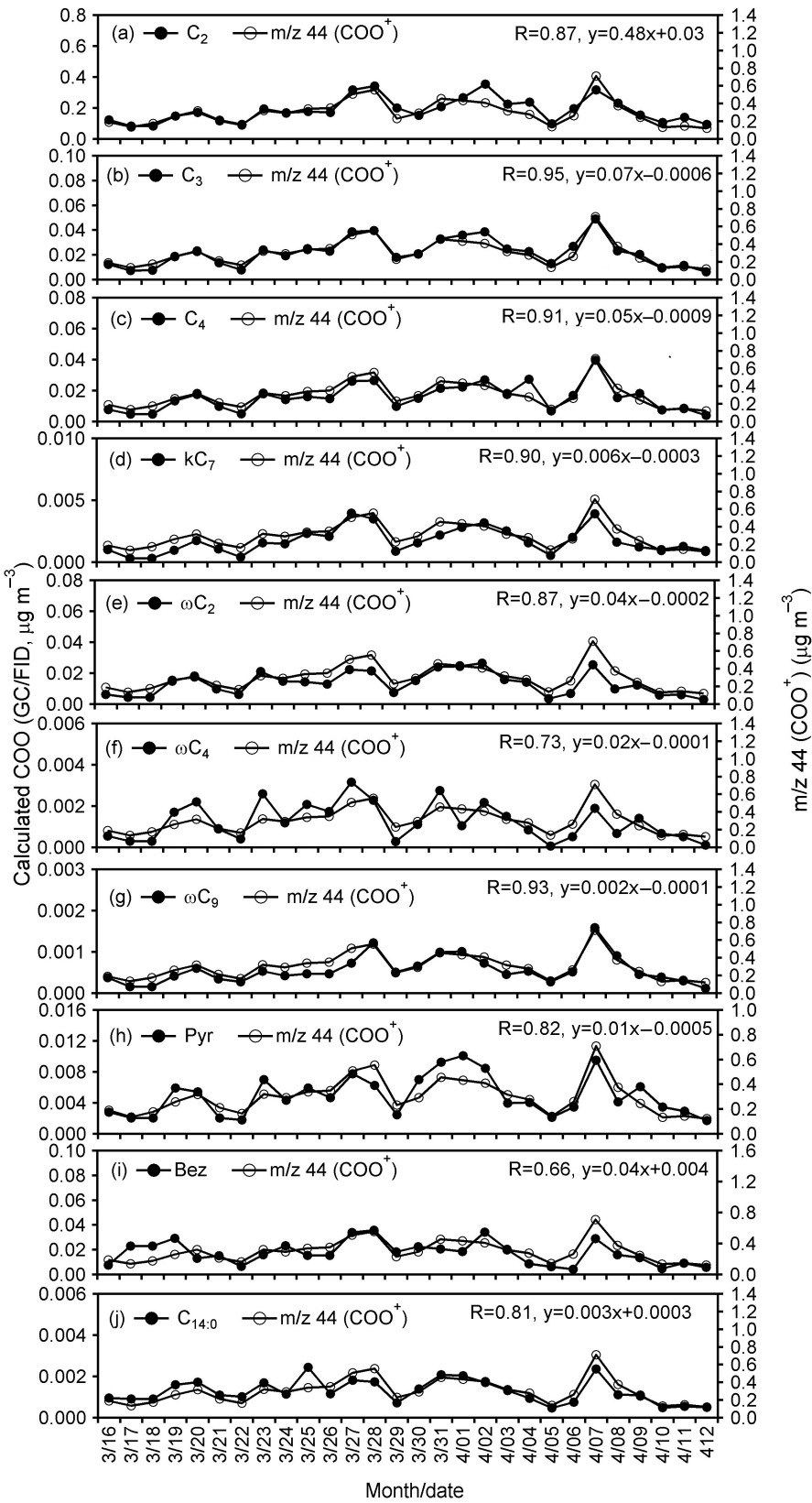

**Figure 4.** Temporal variations in the COO of selected carboxylic acids obtained by GC/FID and m/z 44 (COO$^+$) signal obtained by AMS (**a**) oxalic acid (C$_2$), (**b**) malonic acid (C$_3$), (**c**) succinic acid (C$_4$), (**d**) 4−ketopimelic acid (kC$_7$), (**e**) glyoxylic acid ($\omega$C$_2$), (**f**) 4−oxobutanoic acid ($\omega$C$_4$), (**g**) 9−oxononanoic acid ($\omega$C$_9$), (**h**) pyruvic acid (Pyr), (**i**) benzoic acid (Bez), and (**j**) C$_{14:0}$ fatty acid.

We found strong correlations between m/z 44 signals and the COO of carboxylic acids such as $C_2$ (R = 0.87), $C_3$ (0.95), $C_4$ (0.91), $kC_7$ (0.90), $\omega C_2$ (0.87), $\omega C_4$ (0.73), $\omega C_9$ (0.93), benzoic acid (0.66), and $C_{14:0}$ fatty acid (0.81) (Figure 4). These carboxylic acids contribute to OOA, as presented by AMS m/z 44 in the aerosols [40,45,57,58]. Except for $C_{13:0}$ fatty acid (R = 0.84), the correlation of m/z 44 with other fatty acids is not as good as that of $C_{14:0}$. Figure 5 shows temporal variations in the calculated COO for total diacids, oxoacids, benzoic acid, and total fatty acids obtained by GC and AMS−derived m/z 44. The temporal variations in COO are very similar to those of m/z 44 within the accuracy of the methods. For the calculation of COO, we corrected the concentrations of diacids and related compounds using their recoveries (e.g., 90% for oxalic acid). Some difference (e.g., 7 April, Figure 5) may suggest the presence of unidentified organic acids such as formic, acetic, and propanoic acids that have not been determined by the GC method used.

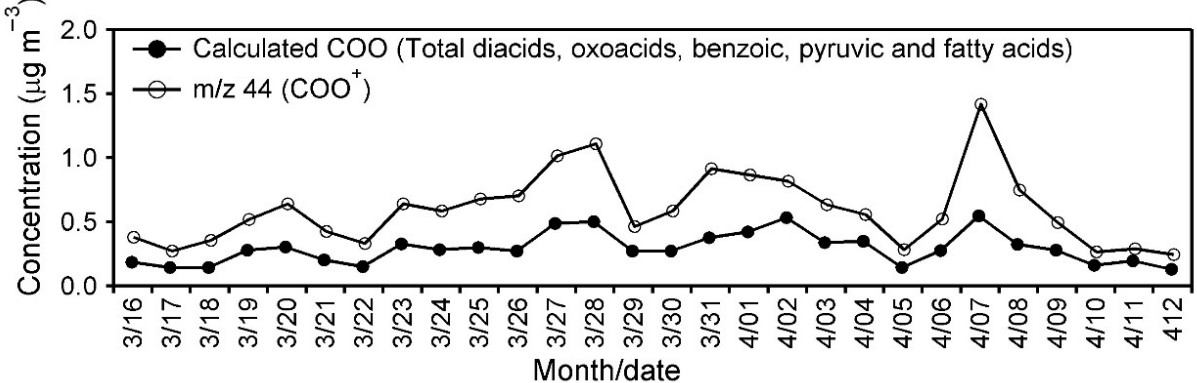

**Figure 5.** Temporal variations in the calculated COO of total diacids, total oxoacids, benzoic acid, and total fatty acids obtained by GC, and m/z 44 ($COO^+$) signals obtained by AMS.

The measurements of the two approaches suggest that GC and AMS techniques are important methods to measure carboxylic acids in aged aerosols ($PM_{1.0}$). The higher concentrations of AMS m/z 44 than COO from GC/FID suggest the presence of other organic acids that were not measured by this GC method. During the period from 29 March to 5 April, m/z 44 signals decreased consistently, while the COO concentrations obtained by off−line GC measurements increased and then decreased. Interestingly, ambient temperature and RH were both lower during this period. Higher temperature and relative humidity (RH) enhanced the oxidation of organics in the atmosphere [82]. Thus, there are other factors that control the COO concentrations of diacids. Some unidentified carboxylic acids may contribute to the m/z 44 signals, which include hydroxy dicarboxylic acids [83,84], mid-chain hydroxy and/or keto carboxylic acids [85–87], and semi-volatile monoacids, including formic and acetic acids in particle phase [27,80,88].

The amounts of COO estimated from diacids and monoacids, including fatty acids, measured by GC show a strong correlation (R = 0.93) with m/z 44 ($COO^+$) signals obtained by AMS (Figure 6). Calculated loadings of m/z 44 ($COO^+$) by AMS and COO (carboxylic acids) by GC agree well, suggesting that carboxylic acids are detected by AMS. This study confirms that m/z 44 signals can be used as reliable markers for oxygenated organic aerosols because major diacids such as oxalic acid are good tracers of secondary organic aerosols (SOA) via photochemical processing in the atmosphere [7,89]. Oxalic acid is not only produced by photochemical processing of organics in daytime, but also produced at night by the oxidation of VOCs with ozone or other oxidants. However, Kawamura et al. (2010) reported that the daytime formation of oxalic acid always overwhelms the nighttime formation. Irei et al. [29] reported a strong correlation between WSOC- and AMS-derived m/z 44 ($COO^+$) signals and suggested that WSOC is enriched with carboxylic acids, although carboxylic acids were not measured.

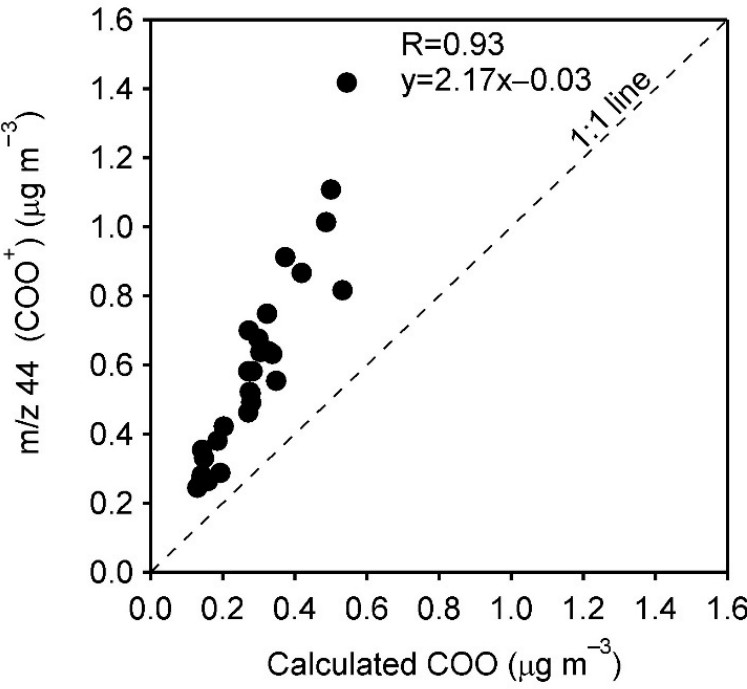

**Figure 6.** Correlation plot for the m/z 44 (COO$^+$) signal measured by AMS and COO calculated for all carboxylic acids (diacids, oxoacids, benzoic acid, and fatty acids) measured by GC.

*3.3. Diacids-Derived COO vs. OM$_{AMS}$ Ratios: Implication for Photochemical Processing during Long-Range Transport to Okinawa*

Oxalic acid is an end product in the oxidation of longer-chain diacids that are mainly formed in the atmosphere by the photochemical processes. Oxalic acid is more abundant in submicron aerosols (0.65–1.1 μm) in Okinawa, suggesting an enhanced photochemical processing of diacids and other precursors in fine mode [82]. Photochemical aging is important in the western North Pacific Rim [75]. These studies demonstrate that the secondary formation of diacids is crucial in Okinawa. AMS−derived m/z 44 signals are dominated by COO$^+$, which mainly originates from the thermal decarboxylation of organic acids [59,64]. Hence, the m/z 44/OM$_{AMS}$ ratio can be used as diagnostic tracer for photochemical processing of organic aerosols [40,59,90]. We found that oxalic acid accounts for $16 \pm 3\%$ of m/z 44 and $3.7 \pm 0.9\%$ of OM$_{AMS}$ in the Okinawa aerosols. This is the first report on the contribution of oxalic acid to OM$_{AMS}$ to discuss the photochemical processing.

Here, we calculated the oxalic−CO−to−OM ratios to better understand the photochemically aged organic aerosols from Okinawa; Okinawa aerosols are seriously aged [48,76]. The oxalic−COO to OM$_{AMS}$ ratios ranged from 0.048–0.11 (av. $0.076 \pm 0.018$) for aged aerosols from Okinawa. Hence, this value should be used for the highly processed aerosols. The average ratios of malonic−COO/OM$_{AMS}$ and succinic−COO/OM$_{AMS}$ are limited to 0.014. For longer-chain diacids, the ratios are still lower than those of malonic and succinic acids. This study shows the COO of specific diacids to OM$_{AMS}$ ratios for the first time in aged aerosols. These ratios are higher when chemical species are more processed. Oxalic acid is produced by the photochemical oxidation of various precursors and is enriched in fine aerosols as an end product of chain reactions of longer-chain diacids and related compounds in the atmosphere [76,91–95]. Several studies have reported higher ratios of oxalic acid to OC when the air mass is transported from urban to downwind remote/rural sites [13,21,49,93,96]. Hence, the diacid–COO/OC ratio can be used as a proxy for photochemical processing.

OA$_{AMS}$ showed strong correlations with the COO of oxalic acid (R = 0.84), malonic acid (0.93), and succinic acid (0.89). Further, we found a strong correlation (R = 0.92) of OA$_{AMS}$

with $SO_4^{2-}$. Sulfate is mainly formed in the aqueous phase by the atmospheric oxidation of $SO_2$. We also found a good correlation between diacids$-COO/Org_{AMS}$ (except for four points) and ambient temperature (R = 0.66). The temperature is always correlated with solar radiation [97]. Hence, diacids$-COO/Org_{AMS}$ can be used as a tracer for photochemical processing in both gaseous and aqueous phases. In general, an OC/EC ratio exceeding a threshold of 2.0 is used to indicate the photochemical production of secondary organic aerosols [98]. Cui et al. [99] reported higher OC/EC ratios for remote and rural sites. The average OC/EC ratios in this study were 3.9. Hence, the secondary formation of organic aerosols is important in Okinawa due to photochemical aging during long-range transport.

As discussed above, m/z 44 /$OM_{AMS}$ ratio can be used as a diagnostic tracer for the photochemical processing of organic aerosols. Takegawa et al. [40] reported that m/z 44/$OM_{AMS}$ ratio < 0.04 is associated with less processed organics; the ratio of 0.04 to 0.08 is related with moderately processed organics; and the ratio > 0.08 is involved with highly processed organics. In this study, we found that in all the samples, m/z 44/$OM_{AMS}$ ratios (range: 0.09–0.13) were greater than 0.08, supporting the notion that organic aerosols from Okinawa were subjected to serious photochemical processing during the atmospheric transport from East Asia to the western Pacific Rim.

Carbonaceous components (OC and EC) have been used to evaluate the secondary formation of organic aerosols, e.g., [100]. Concentrations of OC and EC ranged from 0.41 to 2.49 $\mu g\ m^{-3}$ (av. 1.2 $\mu g\ m^{-3}$) and 0 to 0.85 (0.36 $\mu g\ m^{-3}$), respectively. The OC-to-EC ratio is a useful tool to determine the sources of aerosols (Kunwar and Kawamura, 2014b and references therein). The OC/EC ratios exceeding 2.0 are derived from the enhanced production of SOA [101]. The averaged OC/EC ratio in the present study is ca. 3, suggesting a significant contribution of SOA. Takami et al. [48] reported that low volatility (LV)-OOA is the dominant organics produced by secondary processes over Okinawa, being consistent with the previous results from Cape Hedo [102,103]. Based on the enhanced OC/EC, m/z 44 ($COO^+$)/$OM_{AMS}$ and COO of specific diacid/$OM_{AMS}$ ratios, we consider that the ambient aerosols from Okinawa are photochemically processed during long-range transport from the Asian Continent. In addition, $C_3/C_4$ ratios were used as a proxy of photochemical aging of OA; <1 for primary emissions [12,14,49,86] and >1 for aged aerosols [11]. The average $C_3/C_4$ ratio of 1.3 in this study again suggests a significant photochemical aging of organic aerosols from Okinawa.

We found that the majority of OA is a low-volatility oxygenated organic aerosol (LV−OOA). Our mass spectra showed strong peaks at m/z = 18, 28, and 44, which are typical for organic acids and LV-OOA [40,48,103]. The more oxidized LV−OOA component showed strong correlation with sulfate [104–106]. Further, we can compare our data with those from other locations of the world. Figure S2 in the supporting information shows the relation between $f_{44}$ (ratio of m/z 44 to total organics) and $f_{43}$ (ratio of m/z 43 to total organics). $f_{44}$ axis in Figure S2 can be considered as an indicator of photochemical aging, which leads to an increase in $f_{44}$ [59,107–109]. As seen in Figure S2, $f_{43}$ values are lower than those from Ng et al. [110], suggesting that non-acid oxygenates are less abundant in Okinawa aerosols.

The m/z 44/OA ($f_{44}$) and m/z43/OA ($f_{43}$) ratios depend on the molecular structure of the precursors and types of oxidants [110–112]). The oxidation of aromatic hydrocarbons results in ring-opening [111,113–115]. Sato et al. [111] reported that the $f_{43}$ of benzene oxidation products was very low (0.01), and $f_{44}$ decreased with increasing numbers of alkyl substituents on aromatic rings. The $f_{43}$ and $f_{44}$ values of benzene-derived SOA are 0.016 and 0.17, respectively. The $f_{43}$ and $f_{44}$ values of ethylbenzene products (0.023 and 0.12) [111] during photooxidation are close to those in the present study (0.04 and 0.12). These results are consistent with the hypothesis that aromatic hydrocarbons are a potential source of SOA formation in Okinawa. Docherty et al. [112] reported $f_{44}$ and $f_{43}$ values for naphthalene (range: 0.11–0.17 and 0.015–0.094) and butadiene (range: 0.10–0.14 and 0.03–0.05). The $f_{44}$ and $f_{43}$ values from this study are within those for SOA products from



naphthalene and butadiene [112]. Hence, main the SOA precursors of Okinawa aerosols may be alkyl benzenes, naphthalene, and butadiene via long-range transport (Figure S2).

## 4. Conclusions

We compared the abundances of COO obtained by a capillary GC for carboxylic acids including diacids, oxoacids, benzoic acid, and fatty acids with those of $COO^+$ (m/z = 44) obtained by Aerodyne AMS for the ambient aerosols ($PM_{1.0}$) from Cape Hedo, Okinawa Island, an outflow region from the Asian continent. We found a strong correlation between the two parameters (i.e., carboxylic acids and AMS m/z 44). Although AMS cannot specify the types of carboxylic acids and their molecular species, this study confirms that AMS m/z 44 signals can be used as a surrogate tracer of atmospheric carboxylic acids whose chemical analyses on molecular levels require lots of work and time. However, the detected carboxylic acids with the predominance of oxalic acid account for ca. 50% of AMS m/z 44 signals. This finding further suggests that, in addition to diacids and related compounds, there are significant amounts of organic acids in the $PM_{1.0}$ samples. We suppose that LMW monoacids dominated by formic, acetic, and propanoic acids are the important candidates. We also found higher ratios of OC/EC, m/z 44 $(COO^+)/OM_{AMS}$, and diacid specific-$COO/OM_{AMS}$ in this study, demonstrating that the Cape Hedo aerosols are photochemically highly aged during long-range atmospheric transport from East Asia to the western North Pacific Rim.

**Supplementary Materials:** The following supporting information can be downloaded at: https://www.mdpi.com/article/10.3390/app12168017/s1, Figure S1: Real time variations of m/z 44 and organics measured by AMS in aerosols from Cape Hedo Okinawa. Figure S2: f44 vs. f43 for all the OA component from Okinawa. Except for the red filled circle, all data are adopted from Ng et al. (2010). Table S1: AMS inlet relative humidity, inlet temperature, molar ratios of ammonium to sulphate and mass ratios of organics to sulphate during 2005, 2006 and 2008 in Cape Hedo, Okinawa. Table S2: Concentrations of organic and inorganic species measured by different methods during the same study period. Text S1: Introduces the calculation of specific diacid to m/z 44. References [110–112] are cited in the Supplementary Materials.

**Author Contributions:** Conceptualization, K.K.; methodology, B.K.; software, B.K.; validation, K.K., and B.K.; formal analysis, K.T. and A.T.; investigation, K.K., S.G.A. and B.K.; resources, K.K.; data curation, K.K., and B.K.; writing—original draft preparation, B.K.; writing—review and editing, B.K., and K.K.; visualization, B.K.; supervision, K.K.; project administration, K.K.; funding acquisition, K.K. All authors have read and agreed to the published version of the manuscript.

**Funding:** This study was in part supported by the Japan Society for the Promotion of Science (JSPS) (Grant-in-Aid Nos. 1920405 and 24221001); the Environment Research and Technology Development Fund (B-0903, 2-1403) from the Ministry of the Environment, Japan; and the JSPS Joint Research Program implemented in association with DFG (JRPs-LEAD with DFG:JPJSJRP 20181601).

**Data Availability Statement:** Data can be provided upon the request to the corresponding author or BK.

**Acknowledgments:** We thank K. Okuzawa, Y. Kitamori, and M. Mochida for their help in aerosol sampling at Cape Hedo. We also thank the NOAA Air Resources Laboratory (A.R.L.) for the provision of the HYSPLIT transport model and READY website (http://www.arl.noaa.gov/ready.php, accessed on 18 July 2022) used in this study.

**Conflicts of Interest:** The authors declare no conflict of interest.

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
