# Peer review of "Comparisons of GC-Measured Carboxylic Acids and AMS m/z 44 Signals: Contributions of Organic Acids to m/z 44 Signals in Remote Aerosols from Okinawa Island"

_applsci, doi:10.3390/app12168017_

Round 1

Author Response

Authors’ Responses to Reviewer # 1

July 31, 2022

Reviewer's Comments:

This article has presented the -Comparisons of GC-measured carboxylic acids and AMS m/z 44 signals: Contributions of organic acids to m/z 44 signals in remote aerosols from Okinawa Island (this is also the title of the article by Kunwar et al. 2022). The article presented the various types of acids, such as carboxylic acids, diacids, oxoacids, and fatty acids, collected from Cape Hedo, Okinawa Island. The author used the butyl ester derivatization for offline GC determination. I am convinced that this manuscript should be published after considering some minor suggestions. The English language and style in this manuscript are considerable, however, recommended some sentence structure in few lines.

My minor suggestions and recommendations are:

  1. The abstract should be more concise of results data rather than just expressing the

possibility of obtaining the result. For example, II suggest writing the research findings to show how it can suggest for deriving the remaining signals from other organic acids as presented in line 26 to 28, and also, I recommend adding that what findings of the article confirms that AMS-derived 28 m/z 44 can be used as a surrogate tracer of carboxylicacids how this study confirms.

Response:  Most of the m/z 44 signal is coming from COO in AMS. Dicarboxylic acids and related compounds degraded to give COO (m/z 44), CO (m/z 28) and H2O (m/z 18) in AMS. This kind of fragmentation of dicarboxylic acid is the common phenomenon in AMS. We measured diacids, oxoacids, and fatty acids using GC-FID. Unfortunately, we do not have the data of monoacids such as formate and acetate. However, formate and acetate are also present abundantly in the atmosphere (Mochizuki et al., 2017, 2019). Recently, Chen et al. (2021) reported a significant amount of low molecular weight monocarboxylic acids using high resolution Time-of-Flight AMS. So, the contribution of formate and acetae to m/z 44 may also be significant. There is strong correlation between COO measured by AMS and GC-FID. So, m/z 44 can be used as a surrogate tracer of carboxylic acids. We already mentioned this point in the abstract. Please see line 31 and Figure 4 in the revised MS.

  1. Lines 82 to 86 can be split into two sentences such as- Aerosol samples (PM1.0, n=28) were collected during a field campaign from 17 March 2008 to 13 April 2008 using low volume air sampler and pre-combusted (450°C, 4 hours) quartz fiber filters (Pallflex 2500QAT, 47 mm in diameter). The sampler was installed on the rooftop of the firststory building of Cape Hedo Atmosphere and Aerosol Monitoring Station (CHAAMS, 26º 9' N, 128° 2' E) (Takami et al., 2007), which is located at Cape Hedo in the northwest coast of Okinawa Island, Japan, an outflow region of East Asia (Figure 1).

Response: Following the comment, we split the lines 82-86 into two sentences. Please see lines 85-90 in revised manuscript.

  1. Line 89: …. −20°C to avoid any microbial degradation of organics can be written as

−20°C to avoid microbial organics degradation.

Response: We modified. Please see lines 93-94.

Reviewer 2 Report

The authors investigated the role of carboxylic acids in determining the m/z 44 signals in atmospheric aerosols measured by the AMS. They found that these classes of organic compounds largely contributed to the m/z 44 signal and can be used as an indictor to demonstrate the extent of photochemical activities in the investigated regions. The paper is well written and the results are presented in a clear and concise way. I have a few general comments 

1) The authors shall discussed the potential sampling artifacts of these compounds given each sample were collected for 24 hours. In addition, the samples have been collected and stored since 2008. Would there be any potential issues?

2)An Q-AMS was used for the campaign. What is the resolution of the ions? Do all ions measured at m/z 44 belong to COO+? I understand the authors have discussed this issue in section 3.2. Any evidence for their claims?

 3.In line 267, can the author elaborate why oxalic-COO to OMAMS ratios of 0.02 - 0.05 can be considered as highly processed aerosols? 

4. why the authors claim the major of OA is LV-OOA in line 310?

5. Can the f43 and f44 ratios be well explained by the detection of the carboxylic acids? 

Author Response

Authors’ Responses to Reviewer #2

The authors investigated the role of carboxylic acids in determining the m/z 44 signals in atmospheric aerosols measured by the AMS. They found that these classes of organic compounds largely contributed to the m/z 44 signal and can be used as an indictor to demonstrate the extent of photochemical activities in the investigated regions. The paper is well written and the results are presented in a clear and concise way. I have a few general comments 

  1. The authors shall discussed the potential sampling artifacts of these compounds given each sample were collected for 24 hours. In addition, the samples have been collected and stored since 2008. Would there be any potential issues?

Response: We have added few lines about the sampling artifact. Please see lines 94-96 in the revised MS.

  1. An Q-AMS was used for the campaign. What is the resolution of the ions? Do all ions measured at m/z 44 belong to COO+? I understand the authors have discussed this issue in section 3.2. Any evidence for their claims?

Response: A quadruple mass spectrometer is utilized to analyze the positive ions for unit mass-to-charge (m/z) ratios with the resolution of 200. The m/z 44 peak is the largest in AMS mass spectrum and there may be some contribution of C3H8+ and C2H4O+  to the m/z 44 signals (Yatavelli et al., 2015). The strong correlation between COO measured by AMS and GC-FID showed the strong evidence that m/z 44 belongs COO. Please see Figure 4. We have added few lines about the mass resolution. Please see line 115-116.

  1. In line 267, can the author elaborate why oxalic-COO to OMAMS ratios of 0.02 - 0.05 can be considered as highly processed aerosols?

Response: Our previous study (Kunwar et al., 2014, 2016, 2017) and several other studies (Takami et al., 2007) showed that Okinawa aerosol is more processed. Oxalic acid is formed in the atmosphere by the secondary formation from its precursors during long range atmospheric transport. Hence, this ratio can be used to discuss for the highly processed aerosols. The AMS mass spectra also showed that the highly processed aerosols in Okinawa. We cannot find this kind of ratios in the literature. Both AMS and GC-FID measurements showed that the aerosol from Okinawa is highly processed. Hence, this ratio is used as for the highly processed aerosols. Please see lines 281-293 in the revised MS.

  1. why the authors claim the major of OA is LV-OOA in line 310?

Response: We have added few lines. Please see lines 337-338 in the revised MS.

  1. Can the f43 and f44 ratios be well explained by the detection of the carboxylic acids? 

Response: Yes, we can utilize the f44/f43 ratios to evaluate the contribution of carboxylic acids. The f44/f43 ratios depend on various parameters such as the molecular structure of precursors, types of oxidants and formation processes. f44 is the ratio of  the m/z44/OA. m/z 44 comes from the COO. Hence, the presence of carboxylic acid well explained the f44/f43 ratio. See lines 341-358 in the revised MS.
